# Predicting death by the loss of intestinal function

**Kathreen Bitner** [1]*, **Parvin Shahrestani**[2], **Evan Pardue**[1], **Laurence D. Mueller**[1]

**1** Department of Ecology and Evolutionary Biology, University of California, Irvine, California, United States of America, **2** Department of Biology, California State University, Fullerton, California, United States of America

* kbitner@uci.edu

**Data Availability Statement:** Data are available via DRYAD: https://doi.org/10.7280/D1RQ4W.

**Funding:** All the funding for this study came from the University of California Regents. The funders had no role in study design, data collection and

## Abstract

The ability to predict when an individual will die can be extremely useful for many research problems in aging. A technique for predicting death in the model organism, *Drosophila melanogaster*, has been proposed which relies on an increase in the permeability of the fly intestinal system, allowing dyes from the diet to permeate the body of the fly shortly before death. In this study we sought to verify this claim in a large cohort study using different populations of *D. melanogaster* and different dyes. We found that only about 50% of the individuals showed a visible distribution of dye before death. This number did not vary substantially with the dye used. Most flies that did turn a blue color before death did so within 24 hours of death. There was also a measurable effect of the dye on the fly mean longevity. These results would tend to limit the utility of this method depending on the application the method was intended for.

## Introduction

Evolutionary biologists recognize three phases of adult life in organisms that reproduce multiple times. The first phase occurs prior to reproduction and can be called development. During this phase we expect natural selection to oppose any genetically based reductions in survival since death at these ages means zero fitness. In the second phase, called aging, the strength of natural selection declines with age as first outlined by Hamilton [1]. Under protected conditions, we typically see an age-dependent increase in mortality and a decline in fertility [2]. Finally, at advanced ages organisms enter late-life [3–6]. Again, under protected conditions late-life is characterized by a plateau in age-specific mortality [3–5, 7] female fecundity [6, 8], male virility [9], and age related motor performance decline and specific late-life motor disabilities [10].

Recently we have suggested there is a fourth stage of adult life called the death spiral [6, 8, 11]. The death spiral is a short period prior to death that is marked by a dramatic decline in physiological health. There is evidence of this decline in fecundity [8, 12, 13], supine behavior [14], activity and desiccation resistance [15], and male virility [9]. We have previously shown that the decline in fecundity can be used to predict death [11]. We are looking for a better and faster way to predict death as current phenotypic methods are cumbersome.

analysis, decision to publish, or preparation of the manuscript.

**Competing interests:** The authors have declared that no competing interests exist.

Additional study of the death spiral and a more detailed understanding of the physiological systems that are under decline could be done if there was a reliable and easy way to identify individuals that were about to die. This would permit one to do destructive assays on individuals in the death spiral such as gene expression studies and compare them to similarly aged individuals who are not about to die. Rera et al. [16] describes such a process for *Drosophila melanogaster*. According to [16] and [17] individuals fed food with a blue dye (FD&C blue dye #1) will maintain their ability to prevent the dye from permeating the intestinal barrier until a few days before death. At that time the entire body of the fly will become blue, leading [16] to identify the individuals so colored as "Smurfs" [18–23]

This technique, in principle, offers exactly the assay needed for more detailed analysis of the death spiral. Unfortunately, prior work with the technique has not laid out any detailed analysis of the demographic features of the Smurf phenotype. For instance, what is the average and distribution of the time interval between becoming a Smurf and death? Do these properties change with chronological age? In large samples what fraction of flies become Smurfs prior to death? How does the appearance of the Smurf phenotype vary with other dyes and different populations of *D. melanogaster*? The goal of this study is to answer these questions.

## Methods

### Populations

Five large independent populations of *Drosophila melanogaster* were used in this experiment. Two of these populations, ACO and CO, are large, outbred populations that have been maintained on different age-at-reproduction schedules for hundreds of generations. The ACO population was maintained on 9 day discrete generation cycles. The CO population was kept on 28 day discrete generation cycles. The remaining populations, S93, A4 3852 and Canton S (CAS), were inbred lines raised on three week cycles in the Long lab at the University of California, Irvine. All populations were raised in identical conditions of temperature, food, cultures and density for three generations prior to these experiments.

### Mortality assay

Adult, 14 day old (from egg) flies were knocked out with $CO_2$ gas and placed into individual plastic straws about 4 inches in length and capped with plastic pipette tips on both ends (Fig 1). During anesthetization, a steady supply of $CO_2$ was flowing through a semi-porous plate. The flies were placed on the plate and separated by gender and each fly was gently swept into the plastic straw using a fine painters brush. An equal number of females and males were used per population. Food was provided to each fly at one end of the straw. Each fly was transferred to a new straw with new food and new pipette tips every 3 days to maintain a clean environment. The straw length and girth permitted individuals to fly from one end to the other.

The process of transferring the flies, as well as daily checking of the flies, required a light tapping of the fly into the pipette tip. Cohorts of about 56 adult flies, equal numbers of males and females from each of the five populations were exposed to either control food or food with one of six dyes (Table 1) added to their food. Substantial replication was used. Thus, the original dye, SPS Alfachem, was replicated in 5 different populations, and each population was replicated in 6 different dye environments. The use of different FDA FD&C Blue dye #1's permitted us to determine if the development of the SMURF phenotype was sensitive to the particular dye used. By using a combination of different populations of *D. melanogaster*, which varied in levels of inbreeding, we could determine if the development of the SMURF phenotype was limited to inbred populations.

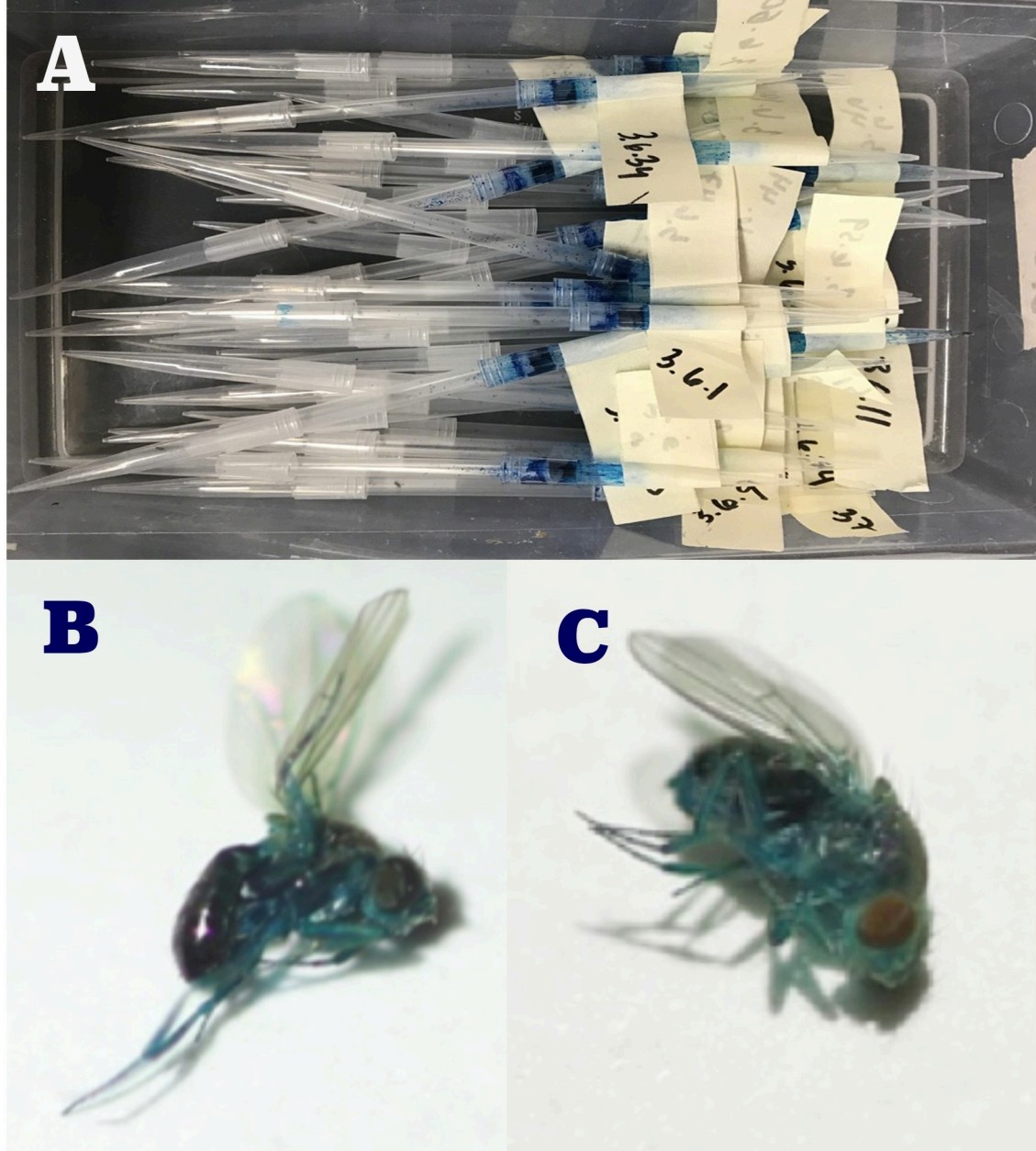

**Fig 1. A.** Adult, 14 day old (from egg) flies were placed into individual plastic straws about 4 inches in length and capped with plastic pipette tips on both ends. Each straw was labeled with a number that allowed us to keep track of each fly, **B.** Blue male Smurf at the time of death, C. Blue female Smurf at time of death. The head, thorax and abdomen have all visibly become blue for both the male and female *D. melanogaster* flies.

The flies were exposed to the blue dyes from day 14 (from egg) continuously to their death. Each fly was individually checked underneath a microscope and light to see if it had become a 'smurf'. Smurf status required that the entire body changed to any variation of a blue color. This was an important distinction as all the *Drosophila* flies fed food with a blue dye would have visible blue coloring in only the gut portion when they weren't a Smurf. Some of the dyes resulted in a slight variation in blue color in the Smurfs. Every day under a microscope with a light we looked for any change of color in the fly thorax, head and abdomen. If the fly was any

**Table 1. The number of flies used in the experiment per population of *D. melanogaster as well* as the total number and sex (M = Male, F = Female) of flies exposed to each dye.** The left column has the five populations used: ACO, CO, S93, A4 3852 and Canton S, and the top row has the medium that the flies were fed: either the control banana molasses food or the banana molasses food with the indicated dye. This table excludes one CO individual in dye 1 whose sex was unknown.

| Population | Control | | Dye 1 | | Dye 2 | | Dye 3 | | Dye 4 | | Dye 5 | | Dye 6 | | Total Flies per Population |
|---|---|---|---|---|---|---|---|---|---|---|---|---|---|---|---|
| | Regular Banana Molasses Food | | Food & SPS Alfachem Blue | | Food & Sigma Aldrich | | Food & Spectrum Blue | | Food & Flavors & Color Blue | | Food & Chemistry Connection | | Food & Electric Blue | | |
| **1: ACO** | Total Flies | | Total Flies | | Total Flies | | Total Flies | | Total Flies | | Total Flies | | Total Flies | | 382 |
| | 57 | | 56 | | 52 | | 54 | | 55 | | 55 | | 53 | | |
| | M | F | M | F | M | F | M | F | M | F | M | F | M | F | |
| | 30 | 27 | 28 | 28 | 27 | 25 | 27 | 27 | 29 | 26 | 27 | 28 | 28 | 25 | |
| **2: CO** | Total Flies | | Total Flies | | Total Flies | | Total Flies | | Total Flies | | Total Flies | | Total Flies | | 395 |
| | 54 | | 54 | | 56 | | 56 | | 57 | | 61 | | 57 | | |
| | M | F | M | F | M | F | M | F | M | F | M | F | M | F | |
| | 27 | 27 | 25 | 29 | 29 | 27 | 27 | 29 | 29 | 28 | 29 | 32 | 28 | 29 | |
| **3: S93** | Total Flies | | Total Flies | | Total Flies | | Total Flies | | Total Flies | | Total Flies | | Total Flies | | 410 |
| | 57 | | 58 | | 59 | | 58 | | 59 | | 60 | | 59 | | |
| | M | F | M | F | M | F | M | F | M | F | M | F | M | F | |
| | 27 | 30 | 29 | 29 | 30 | 29 | 27 | 31 | 30 | 29 | 30 | 30 | 31 | 28 | |
| **4: A4 3852** | Total Flies | | Total Flies | | Total Flies | | Total Flies | | Total Flies | | Total Flies | | Total Flies | | 396 |
| | 57 | | 57 | | 56 | | 57 | | 58 | | 56 | | 55 | | |
| | M | F | M | F | M | F | M | F | M | F | M | F | M | F | |
| | 28 | 29 | 30 | 27 | 27 | 29 | 29 | 28 | 29 | 29 | 27 | 29 | 27 | 28 | |
| **5: Canton S (CAS)** | Total Flies | | Total Flies | | Total Flies | | Total Flies | | Total Flies | | Total Flies | | Total Flies | | 398 |
| | 55 | | 58 | | 58 | | 55 | | 56 | | 57 | | 59 | | |
| | M | F | M | F | M | F | M | F | M | F | M | F | M | F | |
| | 27 | 28 | 29 | 29 | 29 | 29 | 24 | 31 | 28 | 28 | 28 | 29 | 29 | 30 | |
| **Total Flies per dye** | **280** | | **283** | | **281** | | **280** | | **285** | | **289** | | **283** | | **1982** |

shade of blue in all three sections, it was marked as a Smurf and was then checked daily to see when it died. We did not limit our observations to individual sections of the fly, such as only the thorax, for our evaluation of when a fly became a Smurf.

## Tapping

We did the tapping experiment to see if the physical disruption, the process of tapping the fly into the pipette tip, affected the mean longevity and lifespan of the fly. A total of 164 ACO flies were chosen for this assay– 83 males and 81 females. The 164 flies were placed into regular food straws with no dye. A total of 84 flies (42 male and 42 female) were tapped 5 times daily, mimicking the checking that occurred in the original experiment, and the other 81 (41 males and 39 females) flies were not tapped. The flies were transferred to new straws, with fresh food and new pipette tips every 3 days. Each fly was checked daily for movement and if no movement was detected, the fly was classified as deceased on that day. Only ACO flies were used as the purpose of the Tapping experiment was to see if our methods for checking for Smurf flies would affect the mean longevity of the fly.

## Food & dyes

Flies were provided with banana-molasses food with one of the dyes added. The control flies received only banana molasses food in their respective straws. The recipe for the banana

molasses food used in the lab, as well as the experiment, can be found in the Supplemental Portion. Food with dye was prepared by mixing 2.5 grams of each dye to create a 100 ml solution of the banana molasses food mixed with the dye (2.5% wt/vol). Food was always prepared the day before it was needed and stored in a refrigerator until it was used. The dyes were kept separate and carefully handled so no cross-contamination occurred during the preparation and food blending process.

### Statistical analysis

To analyze the effects of dye, sex and population on longevity we let $y_{ijkl}$ be the age at death of the $l$th individual of sex-$i$ ($i$ = 1 (female), 2 (male)), treatment-$j$ ($j$ = 1,..,7 (see Table 1, 7 = control)), and population-$k$ ($k$ = 1,..,5 (see Table 1)). Then a linear model for longevity is,

$$y_{ijkl} = \alpha + \delta_i\beta + \delta_j\gamma_j + \delta_k\pi_k + \delta_j\delta_k\theta_{jk} + \varepsilon_{ijkl}, \tag{1}$$

where $\delta_s$ = 0 if $s$ = 1, and 1 otherwise, $\varepsilon_{ijkl}$ is an error term assumed to have normal distribution with mean 0 and variance $\sigma^2$. An initial test showed no significant differences between sexes so the final model tested did not include the $\beta$ parameter. These tests were run with R (version 3.4.3, R Core team, 2017) and the *lm* function. Pairwise tests with Bonferroni corrections for simultaneous tests were conducted with the R *emmeans* function.

At the time of death each fly was classified according to their sex, population, treatment, and Smurf status (blue: yes or no). Using hierarchical log-linear models (*loglm* function in the R MASS package) we tested in succession whether sex, treatment, and population would have an effect on Smurf status at the time of death.

A $t$-test was performed on the Tapping Experiment results, comparing the mean longevity of the tapped flies versus the non-tapped flies to see if the mechanical disruption would affect their mean longevity.

## Results

We tested the difference in mean longevity for each population in the control environment vs. each of six dyes yielding a total of 30 hypothesis tests (S1 Table, S1 Fig). We found the control populations lived longer in all cases and 7 out of 30 of these tests were significantly different (using a Bonferroni correction for multiple testing). The log-rank tests detected 9 significant differences (S2 Table). Median longevity and maximum longevity were also calculated for each population-dye combination (S3 and S4 Tables). Averaged over the six dye- treatments there were significant differences in mean longevity between all populations and their controls except CO and s93 ($p$ = 0.16, Bonferroni corrected for 5 different tests). Averaged over the five different populations, the control treatments lived significantly longer than every other dye treatment. The controls lived from 4.9 to 9.8 days longer, depending on which dye they were compared to, or about 8% to 17% of the control fly mean longevity (Fig 2).

All FD&C blue dyes did show a Smurf phenotype, though the fraction of flies becoming Smurfs varied considerably from dye to dye, as well as among populations (Fig 3 and S5 Table). These results are consistent with a preliminary study we conducted on 172 ACO flies (S1 Fig).

The null loglinear model with no interactions was compared to a model with an interaction between sex and Smurf status and showed that sex has a significant effect on whether a fly becomes a Smurf ($\chi_1^2 = 30, p < 10^{-5}$). Averaged over all blue dye treatment populations, 22% of the males became Smurfs and 34% of the females became Smurfs. If we add an interaction between dye treatment and Smurf status to the previous model with the sex interaction there is a significant effect of dye treatment ($\chi_5^2 = 13.0, p < 0.022$). Finally, adding an interaction

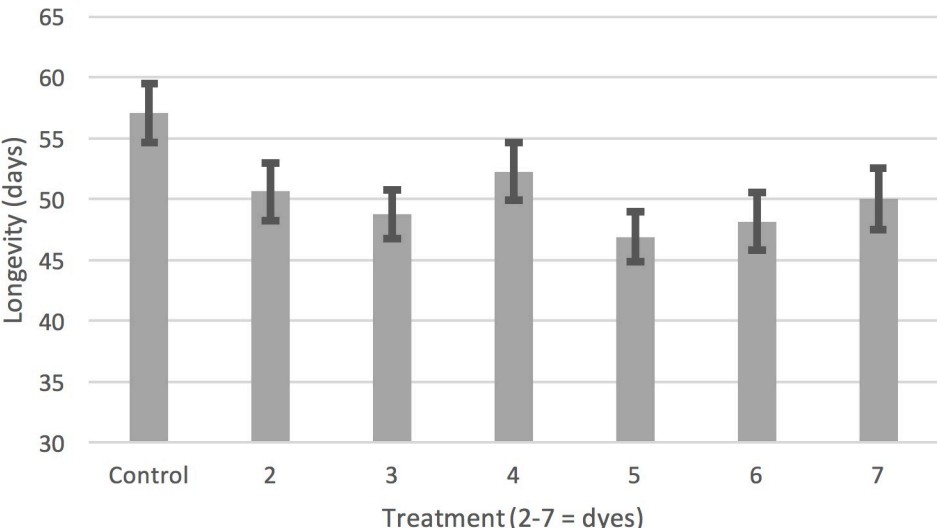

**Fig 2. The mean longevity of five populations of *D. melanogaster* in the control and six different dye treatments.**
Bars are 95% confidence intervals calculated from a pooled variance estimate by the *emmeans* R function. Each dye treatment resulted in a significant reduction in mean longevity compared to the control treatment. Treatment 2: SPS Alfachem Blue, 3: Sigma Aldrich, 4: Spectrum Blue, 5: Flavors and Color Blue, 6: Chemistry Connection Blue, 7: Electric Blue.

between population and Smurf status to the previous model with interactions between sex, treatment and Smurf status there is a significant effect of population ($\chi^2_4 = 28$, $p = 0.00001$). Thus, achieving the Smurf phenotype before death is significantly affected by sex, dye, and population. However, the majority of flies never showed the Smurf phenotype prior to death.

The flies became a distinct Blue color in their abdomen, thorax and head when they became a 'Smurf' (Fig 1). This could be seen in some flies as much as 3 or 4 days before their death.

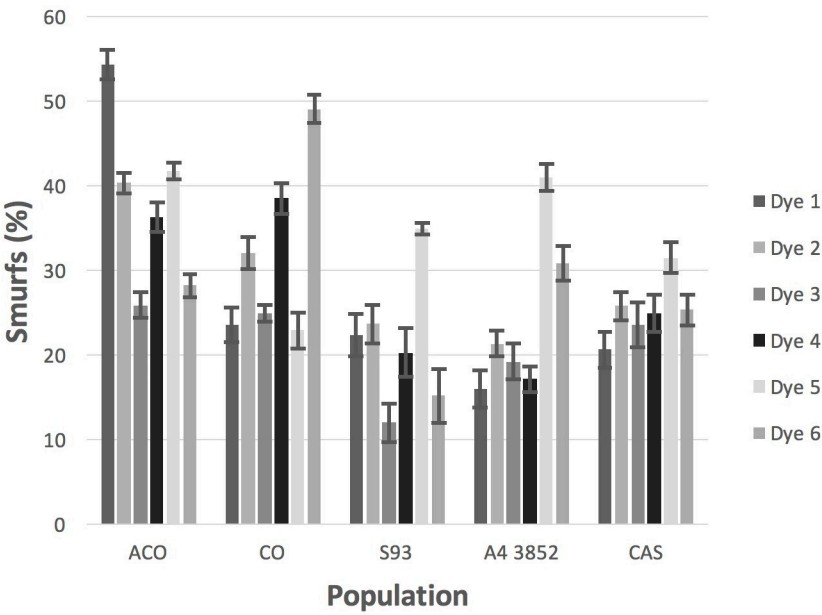

**Fig 3. Percentage of flies that became Smurfs for each population and dye.** Bars are standard errors.

However, of the flies that became a Smurf, the majority did so on the day of their death or one day before death (Fig 4).

A *t*-test was run on the tapping experiment, comparing the mean longevity of the tapped flies versus the non-tapped flies to see if the mechanical disruption would affect their mean longevity. The males were not affected by tapping, with a mean longevity of 53.2 days for those tapped and 53.9 days for those not tapped ($p = 0.83$, Table 2). Likewise females were not affected due to the tapping mechanism either, with a mean longevity of 54.5 days for the tapped females and 54.4 days for the untapped females ($p = 0.99$, Table 2).

## Conclusions

This study has established a number of important conclusions. (1) All dyes used have significant negative effects on mean longevity, with decreases ranging from 5 to almost 10 days. (2) Only a small fraction of the flies show the Smurf phenotype prior to death. Over all populations and dyes 22% of males and 34% of females became Smurfs. (3) Among the small fraction that do become Smurfs most (40–60%) become blue during their last 24 hours of life. Thus, even with daily checks most of the Smurf flies will be dead when initially identified as Smurf making their utility for gene expression studies useless. As can be seen from Table 1, there was substantial replication, with each population undergoing 6 different dyes and a control. The original dye used in Rera et al. [16], SPS Alfachem, was replicated in 5 different populations, and each population was replicated in 6 different dye environments, allowing for substantial replication across the whole experiment. The three results cited above were consistently seen across all the replicates suggesting that these findings are robust.

These results certainly contradict prior claims [16]. Rera et al [16] suggested that essentially all flies become Smurfs prior to death and that the dyes do not affect survival. Certainly, one can claim there were differences in handling or techniques used in these studies [16]. This is challenging to evaluate. We note that the food used in our study has 2.5 grams of dye per 100 mL of food (2.5% wt/vol), which is the same dye concentration that Rera et al [16] put in their food [17]. Thus, our observations of increased mortality due to dye cannot be attributed to overdosing. We also tested whether the tapping employed in our experimental technique could explain the mean longevity differences. That experiment showed no detectable effects of tapping on either male or female longevity. We only tested one population, ACO, for an effect of tapping on mean longevity. Thus, for the ACO population it is clear that dyes are responsible for their reduced longevity not tapping. While it is theoretically possible that the other populations are not affected by the dyes but are affected by tapping, we believe this is an unlikely possibility.

There might be variation in how much food a fly consumes, but since each fly was in their own environment with only the dyed food, the flies had no other option but to consume the food or die from starvation. The purpose of this assay technique is to identify flies about to die under normal husbandry protocols. Future experiments can focus on whether the dead 'non-Smurfed' flies consumed food before death or not. But if the majority of the flies are dying without the distinguishing blue body color, then the technique is of little practical use. Lastly, most flies which did turn blue did so during their last 24 hours of life. This also renders the technique less useful for collecting live flies shortly before their death.

Many experiments have used the Smurf Assay technique [18–23]. However, the widespread use of the Smurf assay to differentiate between aging flies and young flies is not justified. At older ages, less flies Smurfed than flies that were younger. Prior research has demonstrated that the technique will differentiate between individuals that loose intestinal integrity and

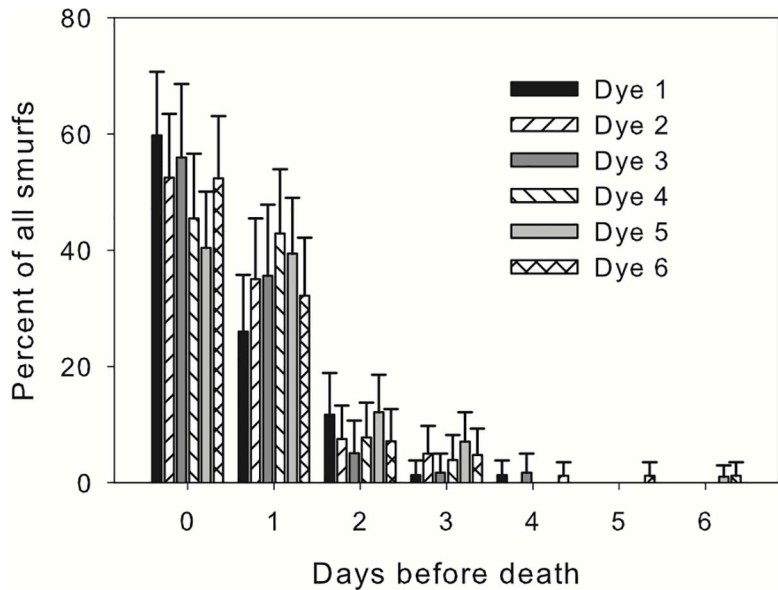

**Fig 4. Timing of SMURF appearance using only flies that satisfied our criteria for being a Smurf.** Most flies appeared to be Smurfs on the day they are found dead or 1 day before death. Of the *D. melanogaster* that became a Smurf, a majority of them did so on the day of their death or one day before. Bars are simultaneous 95% confidence intervals. Dye 1: SPS Alfachem Blue, Dye 2: Sigma Aldrich, Dye 3: Spectrum Blue, Dye 4: Flavors and Color Blue, Dye 5: Chemistry Connection Blue, and Dye 6: Electric Blue.

**Table 2. Mean longevity from the tapping experiment.** The *p*-value is for a *t*-test for different mean longevities of same sex treatments. There is no discernible difference in the mean longevity of the flies that were tapped versus those that were not tapped.

| Sex | Tapping | Mean Longevity (Days) | 95% Confidence Interval | *p* value |
|---|---|---|---|---|
| Male | Yes | 53.2 | (47.8, 58.5) | 0.83 |
| Male | No | 53.9 | (49.4, 58.5) | |
| Female | Yes | 54.5 | (49.0, 60) | 0.88 |
| Female | No | 54.4 | (49.0, 59.8) | |

become Smurfs and those that don't, but they fail to provide exact details on how many individuals become Smurfs prior to death.

Our results also demonstrate significant effects of fly population of origin and dye on both mean longevity and frequency of Smurfs. However, these effects are essentially background noise to the major observations that only about 28% of flies ever become Smurfs and those that do only do so on their day of death or one day before death.

## Supporting information

**S1 File. Standard rose and Mueller lab banana food recipe.**
(DOCX)

**S1 Fig. Mean longevity in days of flies in the control environment and dyes across all 5 populations.** Standard error bars. When pooling the dyes against the control, the control flies lived significantly longer than the flies in an environment with dye in the food. The dyes used: Dye 1: SPS Alfachem Blue, Dye 2: Sigma Aldrich, Dye 3: Spectrum Blue, Dye 4: Flavors and Color Blue, Dye 5: Chemistry Connection Blue, and Dye 6: Electric Blue.
(TIF)

**S2 Fig. Results of the preliminary study of 172 ACO1 adults.** The percent of first appearance of all 81 Smurfs as a function of the days before death when raised on food with dye 1. The bars are simultaneous 95% confidence intervals. The majority became Smurfs on the day they were found dead (day 0) or 1 day before death. A total of 47% (95% confidence interval, (39%, 54%)) eventually became Smurfs. The mean longevity (from egg) of all flies in this experiment was 32.9 days (95% confidence interval ±1.4 days). These results are consistent with those in the full experiment. Specifically, less than 50% of all flies became Smurfs prior to death and those that did become Smurfs most frequently did so on the day or day before they died.
(TIF)

**S1 Table. Average longevity for every population and every dye used in the experiment.** Also provided the average day of Smurfing.
(DOCX)

**S2 Table. Log-rank *p*-values comparing the control treatment to each of the other populations at each of the dye treatments.** These results are from the R *survdiff* function in the *survival* package. The significant results are shown in bold (using the Bonferroni correction for 30 tests).
(DOCX)

**S3 Table. The median longevity for each population at every dye treatment.** The median longevity for each population was greater in the control treatment in than every other dye except for A4 3852 and dye 1.
(DOCX)

**S4 Table. The maximum longevity for each population, dye combination.**
(DOCX)

**S5 Table. The number of flies that Smurfed per population and per dye, as well as the number of total flies per population and dye.**
(DOCX)

## Acknowledgments

A big thank you to the undergraduates in the lab who helped with the experiment.

## Author Contributions

**Conceptualization:** Kathreen Bitner, Parvin Shahrestani, Evan Pardue.

**Data curation:** Kathreen Bitner, Laurence D. Mueller.

**Formal analysis:** Laurence D. Mueller.

**Investigation:** Kathreen Bitner, Evan Pardue.

**Methodology:** Kathreen Bitner, Parvin Shahrestani, Evan Pardue.

**Project administration:** Kathreen Bitner, Parvin Shahrestani, Laurence D. Mueller.

**Resources:** Laurence D. Mueller.

**Software:** Laurence D. Mueller.

**Supervision:** Kathreen Bitner.

**Visualization:** Kathreen Bitner.

**Writing – original draft:** Kathreen Bitner.

**Writing – review & editing:** Kathreen Bitner, Laurence D. Mueller.

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
