## [Editor Report · Decision Letter 0]

5 Nov 2019

PONE-D-19-29500

The Efficacy of Predicting Death by the Loss of Intestinal Function

PLOS ONE

Dear Dr Bitner,

Thank you for submitting your manuscript to PLOS ONE. After careful consideration, we feel that it has merit but does not fully meet PLOS ONE’s publication criteria as it currently stands. Therefore, we invite you to submit a revised version of the manuscript that addresses the points raised during the review process.

Although short, the manuscript reports on issues that impact a good number of labs in the field. However as it stands it is difficult to send for review because of formatting issues.

Please remove the figure legend text from within teh Results section and add it to the end after the references.

Same for the tables and their legends. Please move them to the respective figure and legend sections.

It is useful if a photo of the "food straw" used was provided.

We would appreciate receiving your revised manuscript by Dec 20 2019 11:59PM. To enhance the reproducibility of your results, we recommend that if applicable you deposit your laboratory protocols in protocols.io, where a protocol can be assigned its own identifier (DOI) such that it can be cited independently in the future. For instructions see: http://journals.plos.org/plosone/s/submission-guidelines#loc-laboratory-protocols

We look forward to receiving your revised manuscript.

Kind regards,

Efthimios M. C. Skoulakis, PhD

Academic Editor

PLOS ONE

Journal Requirements:

"No -The funders had no role in study design, data collection and analysis, decision to publish, or preparation of the manuscript.".

Please provide an amended Funding Statement that declares *all* the funding or sources of support received during this specific study (whether external or internal to your organization) as detailed online in our guide for authors at http://journals.plos.org/plosone/s/submit-now.  

Please state what role the funders took in the study.  If any authors received a salary from any of your funders, please state which authors and which funder. If the funders had no role, please state: "The funders had no role in study design, data collection and analysis, decision to publish, or preparation of the manuscript."
---

## [Author Response · Author response to Decision Letter 0]

12 Nov 2019

Thank you for your time to review our paper. Please see below for the original points mentioned in the review, as well as our responses to each. 

1. Although short, the manuscript reports on issues that impact a good number of labs in the field. However as it stands it is difficult to send for review because of formatting issues.

-We have addressed all of the issues and concerns below regarding formatting issues and we are submitting a Revised Manuscript. 

2. Please remove the figure legend text from within teh Results section and add it to the end after the references.

-The figure legend text from within the Results section was moved and placed at the end of the references in a section called Supporting Material. 

3. Same for the tables and their legends. Please move them to the respective figure and legend sections.

-All of the Tables and legends were moved from their current positions to the respective Supporting Material section at the end of the paper after the References, which now contains the tables, their legends, and figure legends. 

4. It is useful if a photo of the "food straw" used was provided.

-A straw photo was inputted into the Revised Manuscript as Figure 1. All other figures were numerically renamed – Figure 1 became Figure 2, Figure 2 became Figure 3, etc. 

Journal Requirements:

-We made sure our paper meets your requirements. All citations were fixed, as well as some minor formatting issues throughout the manuscript. The only part of the paper that is set up differently than your requirements is that our tables, table legends, and figure legends were moved to the end of the paper as was addressed and requested in the original concerns (please see above, points 2 and 3). 

-We will upload the data and have it ready at acceptance. 

"No -The funders had no role in study design, data collection and analysis, decision to publish, or preparation of the manuscript.".

a. Please provide an amended Funding Statement that declares *all* the funding or sources of support received during this specific study (whether external or internal to your organization) as detailed online in our guide for authors at http://journals.plos.org/plosone/s/submit-now.  

b. Please state what role the funders took in the study.  If any authors received a salary from any of your funders, please state which authors and which funder. If the funders had no role, please state: "The funders had no role in study design, data collection and analysis, decision to publish, or preparation of the manuscript."

-Regarding the financial disclosure, we will include the amended statements in the cover letter. 

Reviewers' comments:

-There were no reviewers comments in this section, therefore we didn’t have anything to address and change other than the original comments above.

---

## [Decision Letter · Decision Letter 1]

2 Dec 2019

PONE-D-19-29500R1

The Efficacy of Predicting Death by the Loss of Intestinal Function

PLOS ONE

Dear Dr Bitner,

Thank you for submitting your manuscript to PLOS ONE. After careful consideration, we feel that it has merit but does not fully meet PLOS ONE’s publication criteria as it currently stands. Therefore, we invite you to submit a revised version of the manuscript that addresses the points raised during the review process.

There are a few technical issues that need to be thoroughly addressed as suggested by both reviewers. In fact the experimental protocol and methodological questions of reviewer 2 and the issues raised regarding replication have to be addressed thoroughly with new experiments if necessary.

We would appreciate receiving your revised manuscript by Jan 16 2020 11:59PM. To enhance the reproducibility of your results, we recommend that if applicable you deposit your laboratory protocols in protocols.io, where a protocol can be assigned its own identifier (DOI) such that it can be cited independently in the future. For instructions see: http://journals.plos.org/plosone/s/submission-guidelines#loc-laboratory-protocols

We look forward to receiving your revised manuscript.

Kind regards,

Efthimios M. C. Skoulakis, PhD

Academic Editor

PLOS ONE

Reviewers' comments:

Reviewer's Responses to Questions

**Comments to the Author**

1. If the authors have adequately addressed your comments raised in a previous round of review and you feel that this manuscript is now acceptable for publication, you may indicate that here to bypass the “Comments to the Author” section, enter your conflict of interest statement in the “Confidential to Editor” section, and submit your "Accept" recommendation.

Reviewer #1: All comments have been addressed

Reviewer #2: (No Response)

2. Is the manuscript technically sound, and do the data support the conclusions?

Reviewer #1: Yes

Reviewer #2: No

3. Has the statistical analysis been performed appropriately and rigorously? 

Reviewer #1: Yes

Reviewer #2: I Don't Know

4. Have the authors made all data underlying the findings in their manuscript fully available?

Reviewer #1: Yes

Reviewer #2: No

5. Is the manuscript presented in an intelligible fashion and written in standard English?

Reviewer #1: Yes

Reviewer #2: Yes

6. Review Comments to the Author

Reviewer #1: This is a manuscript reporting on the reliability of using dyes as food supplements to predict death in Drosophila. In the original paper: Rera M, Clarck RI, Walker DW.M. 2012. Intestinal barrier dysfunction links metabolic and inflammatory markers of aging to death in Drosophila. Proc Natl Acad Sci U S A. 2012 Dec 26;109(52):21528-33, has been suggested that intestinal barrier dysfunction predicts impending death in individual flies. The underlying hypothesis here is that when the intestine epithelium brakes down during aging then the dye, consumed with the food, diffuses from the gut to the rest of the body via circulation and thus the fly exhibits full coloration; it becomes smurf.

In contrast, the results of the present manuscript clearly demonstrate that this is not the case. By using different dyes at similar to the Rera et al, 2012 paper concentration the authors demonstrate that only a minor percentage of flies exhibit full body dye coloration (smurfs) and mostly one day prior to death or even the day of death. This suggests that the smurf phenotype is not a useful biomarker of death.

The findings of this manuscript, previously published papers by the group and a recent paper Gaitanidis et al, 2019 (Gaitanidis A., Dimitriadou A., Dowse H., Sanyal S. Duch C., Consoulas C. (2019). Pre-death morbidity in Drosophila and its compression. Aging, 11:1850-1873) demonstrate that apart from the gradual decay during aging, fit flies may collapse and die within a very short time (sudden death). The process has been termed the death spiral or sudden death and can be fast. This is exactly demonstrated in the present manuscript. Few flies become smurf several days prior to death whereas others (the majority) only the day of death.

That said, I firmly believe that this manuscript should be published.

Minor comments

203 We

204 do note that the food used in our study has only 2.5 grams of dye per 100 mL of food whereas

205 Rera put 3.12 grams of dye in 100 mL of food [16].

-Please refer to the paper `Rera et al, 2012` than to one author. All authors worked for this publication. Furthermore, in the materials and methods section of the aforementioned paper is clearly mentioned that the dye concentration is 2.5% (wt/vol). Thus, remove or modify your claim.

154 the control treatments lived significantly longer then every other treatment.

-Please correct.

213 integrity and become Smurfs andthose that don’t, but they fail to provide exact details on how

214 many individuals become s Smurfs prior to death.

-Please correct

-Add and refer to the Gaitanidis et al, 2019 paper (Gaitanidis A., Dimitriadou A., Dowse H., Sanyal S. Duch C., Consoulas C. (2019). Pre-death morbidity in Drosophila and its compression. Aging, 11:1850-1873).

-The manuscript needs proof reading.

Reviewer #2: This is a very (too) short manuscript that reports an interesting observation but it is lacking a lot of critical information. It is unclear how the experiments were done and it is impossible to replicate the experiments with the information provided.

The authors do not indicate what is actually measured, stating only longevity. Is it the average, mean, max longevity? It is standard in the field of aging to provide all the data in separate tables (in supplemental data), like for instance Table S1 in PNAS 111(22):8137,2014. It is IMPERATIVE that the authors provide tables containing each experimental replicate for each dye and control with the corresponding sample size, average, median and max longevity, and logrank P values (% smurfs included but a second table would be better to be able to include all daily scorings done). Table 1 and 2 are very confusing and strongly suggest that the experiment have been done only once. The material and methods indicates that cohorts of about 56 flies were used but table 1 indicates for example that ~280 flies were exposed per Dye/Control which means that 56 flies from each population was used and based on the number used for each population in table 2 (for instance ACO) 383 flies is the number of flies needed for one experiment (control + 6 dyes: 56*7=392). It is also suspicious that figure 2 has error bars (no explanation in legend, is it SD, SE?) that are all exactly identical. This article cannot be published and I cannot properly evaluate it until the authors provide complete experimental details and data, and that the experiment has been replicated.

The gender of the animals tested is also not provided but the statistical analysis section of the material and method does indicate that males and females were used.

The authors performed an important control experiment to see if tapping affect longevity yet do not provide any data/figure from these experiments nor it is mentioned in the result, yet the discussion states on line 177 that the experiment show no detectable effects of taping. There is no justification to why the authors perform this test only with ACO females while the subsequent experiments use both gender and 4 additional genotypes.

The authors do not report how the flies were collected (anesthesia or not, kind of anesthesia, duration of anesthesia and time window/age interval).

The authors do not indicate how and if they ensure that each fly has actually consumed the food. From the material and method, I guess that the authors did check coloring of the gut but should clearly indicates that all flies did show a blue gut. It remains possible that the proportion of flies that are not “smurfing” consumed less food and consequently not enough dye is ingested to display color outside of the gut.

The composition of the fly food used is not provided.

Figure 1 and 4 should be combined together (method of delivery/monitoring, example of smurf flies).

7. PLOS authors have the option to publish the peer review history of their article (what does this mean?). If published, this will include your full peer review and any attached files.

Reviewer #1: No

Reviewer #2: Yes: Laurent Seroude

---

## [Author Response · Author response to Decision Letter 1]

8 Jan 2020

Reviewer’s comments will be in standard font and our reply will be in red italic font in the attached word document. Here, our response is simply below the the reviewer's comments. 

REVIEWER #1

This is a manuscript reporting on the reliability of using dyes as food supplements to predict death in Drosophila. In the original paper: Rera M, Clarck RI, Walker DW.M. 2012. Intestinal barrier dysfunction links metabolic and inflammatory markers of aging to death in Drosophila. Proc Natl Acad Sci U S A. 2012 Dec 26;109(52):21528-33, has been suggested that intestinal barrier dysfunction predicts impending death in individual flies. The underlying hypothesis here is that when the intestine epithelium brakes down during aging then the dye, consumed with the food, diffuses from the gut to the rest of the body via circulation and thus the fly exhibits full coloration; it becomes smurf.

In contrast, the results of the present manuscript clearly demonstrate that this is not the case. By using different dyes at similar to the Rera et al, 2012 paper concentration the authors demonstrate that only a minor percentage of flies exhibit full body dye coloration (smurfs) and mostly one day prior to death or even the day of death. This suggests that the smurf phenotype is not a useful biomarker of death.

The findings of this manuscript, previously published papers by the group and a recent paper Gaitanidis et al, 2019 (Gaitanidis A., Dimitriadou A., Dowse H., Sanyal S. Duch C., Consoulas C. (2019). Pre-death morbidity in Drosophila and its compression. Aging, 11:1850-1873) demonstrate that apart from the gradual decay during aging, fit flies may collapse and die within a very short time (sudden death). The process has been termed the death spiral or sudden death and can be fast. This is exactly demonstrated in the present manuscript. Few flies become smurf several days prior to death whereas others (the majority) only the day of death.

That said, I firmly believe that this manuscript should be published.

This is a succinct and accurate summary of the paper.

203 We

204 do note that the food used in our study has only 2.5 grams of dye per 100 mL of food whereas

205 Rera put 3.12 grams of dye in 100 mL of food [16].

-Please refer to the paper `Rera et al, 2012` than to one author. All authors worked for this publication. Furthermore, in the materials and methods section of the aforementioned paper is clearly mentioned that the dye concentration is 2.5% (wt/vol). Thus, remove or modify your claim.

We changed the reference throughout the paper to Rera et al 2012 and removed any reference to only “Rera”. We revised the text to reflect that Rera et al. 2012 used food with a 2.5% dye concentration.

154 the control treatments lived significantly longer then every other treatment.

-Please correct.

English corrected.

213 integrity and become Smurfs andthose that don’t, but they fail to provide exact details on how

214 many individuals become s Smurfs prior to death.

This typo has been corrected.

-Add and refer to the Gaitanidis et al, 2019 paper (Gaitanidis A., Dimitriadou A., Dowse H., Sanyal S. Duch C., Consoulas C. (2019). Pre-death morbidity in Drosophila and its compression. Aging, 11:1850-1873).

This reference was added. Thank you very much for the recommendation

-The manuscript needs proof reading.

We have done more careful proof reading and consequently made numerous additional corrections to the manuscript.

REVIEWER #2

This is a very (too) short manuscript that reports an interesting observation but it is lacking a lot of critical information. It is unclear how the experiments were done and it is impossible to replicate the experiments with the information provided.

We have added expanded discussion of the experimental methods including a detailed recipe for our standard fly food which is included in the appendix. 

[as well as the graphs that were requested have been created and are provided in the supplemental section of the paper. A new table was created that clarifies the number of flies used per population and per dye, showing that the experiment had replication. The original dye used, SPS Alfachem, was replicated in 5 different populations, and each population was replicated in 6 different dye environments, allowing for substantial replication across the whole experiment.]

The authors do not indicate what is actually measured, stating only longevity. Is it the average, mean, max longevity? 

We are measuring mean longevity and this has been clarified in the paper.

It is standard in the field of aging to provide all the data in separate tables (in supplemental data), like for instance Table S1 in PNAS 111(22):8137,2014. 

As indicated on the cover page we will deposit the raw data in DRYAD upon acceptance of the paper. 

It is IMPERATIVE that the authors provide tables containing each experimental replicate for each dye and control with the corresponding sample size, average, median and max longevity, and logrank P values (% smurfs included but a second table would be better to be able to include all daily scorings done).

We now have tables in the main part of the paper or in a supplement which give details for each population and dye combination, sample sizes and mean longevities. We have not included the median, maximum longevity or logrank p-values. We did no analyses of the median or maximum longevity. Likewise, since we do not have censored data, we can explore the effects of population and dye more effectively with a standard linear model which we have done. Certainly, with our raw data available any interested scientist can do additional analyses of our data. 

Table 1 and 2 are very confusing and strongly suggest that the experiment have been done only once. The material and methods indicates that cohorts of about 56 flies were used but table 1 indicates for example that ~280 flies were exposed per Dye/Control which means that 56 flies from each population was used and based on the number used for each population in table 2 (for instance ACO) 383 flies is the number of flies needed for one experiment (control + 6 dyes: 56*7=392). 

We created a new table (Table 1) that clarifies the number of flies used per population and per dye in the experiment, as well as the number of males and females used. As can be seen from the table, there was substantial replication, with each population undergoing 6 different dyes and a control. The original dye used, SPS Alfachem, was replicated in 5 different populations, and each population was replicated in 6 different dye environments, allowing for substantial replication across the whole experiment. Most importantly the major results we reach are consistent across the five different populations and six different dyes. It is likely that repeating these experiments in a different month or year would not yield the exact results. However, the variation in this experiment due to fly populations that vary tremendously in levels of inbreeding and evolutionary history and the use of many different provides a robust testing regime.

It is also suspicious that figure 2 has error bars (no explanation in legend, is it SD, SE?) that are all exactly identical. This article cannot be published and I cannot properly evaluate it until the authors provide complete experimental details and data, and that the experiment has been replicated.

The figure legend now clarifies that the bars are 95% confidence intervals that were estimated by the R function emmeans. This function uses a pooled estimate of variance. Since the degrees of freedom are similar for each dye the confidence intervals look similar in size. This pooling is not unreasonable given the similarity in longevities.

The gender of the animals tested is also not provided but the statistical analysis section of the material and method does indicate that males and females were used.

This has been clarified in multiple places in the paper most prominently in Table 1. There were roughly equal numbers of males and females used in the experiment.

The authors performed an important control experiment to see if tapping affect longevity yet do not provide any data/figure from these experiments nor it is mentioned in the result, yet the discussion states on line 177 that the experiment show no detectable effects of taping. There is no justification to why the authors perform this test only with ACO females while the subsequent experiments use both gender and 4 additional genotypes.

We have clarified that both males and females were used in this experiment. The exact numbers of flies used for the Tapping experiment are now described in the Materials and Methods. The results of the tapping experiment are summarized in a new Table 2 and discussed in the Results section. 

The authors do not report how the flies were collected (anesthesia or not, kind of anesthesia, duration of anesthesia and time window/age interval).

This has been clarified in detail in the methods. The flies were knocked out using CO2, separated by gender and placed into the individual straw containers.

The authors do not indicate how and if they ensure that each fly has actually consumed the food. From the material and method, I guess that the authors did check coloring of the gut but should clearly indicates that all flies did show a blue gut. It remains possible that the proportion of flies that are not “smurfing” consumed less food and consequently not enough dye is ingested to display color outside of the gut.

While it is true we have no way knowing exactly if the flies consumed the food, this technique consisted of putting dye in the food and following body color changes. There might be variation in how much food a fly consumes, but since each fly was in their own environment with only the dyed food, the flies had no other option but to consume the food or die from starvation. This was already clarified in the materials and methods section regarding the fact that all flies did have a blue gut: “Smurf status required that the entire body changed to any variation of a blue color. This was an important distinction as Drosophila flies fed food with a blue dye would have visible blue coloring in only the gut portion when they weren’t a Smurf.” In this paper we do not try to assess why the technique fails, uneven food consumption is one possibility. But at the present time the method relies on the flies consuming the food with dye.

The composition of the fly food used is not provided.

The food used is a banana molasses recipe that has been used in our lab for over 3 decades. The detailed recipe is given in the supplementary material

Figure 1 and 4 should be combined together (method of delivery/monitoring, example of smurf flies).

The two figures are now combined together as requested.

---

## [Decision Letter · Decision Letter 2]

23 Jan 2020

PONE-D-19-29500R2

Predicting Death by the Loss of Intestinal Function

PLOS ONE

Dear Dr Bitner,

Thank you for submitting your manuscript to PLOS ONE. After careful consideration, we feel that it has merit but does not fully meet PLOS ONE’s publication criteria as it currently stands. Therefore, we invite you to submit a revised version of the manuscript that addresses the points raised during the review process.

Although improved, the manuscript needs addition careful attention and thorough response to the comments and suggestions raised by reviewer #2 and pertain to the methodology and reproducibility of the data, both cardinally critical issues for publication. It is essential that they are thoroughly addressed as this would be the third revision of the manuscript! 

We would appreciate receiving your revised manuscript by Mar 08 2020 11:59PM. To enhance the reproducibility of your results, we recommend that if applicable you deposit your laboratory protocols in protocols.io, where a protocol can be assigned its own identifier (DOI) such that it can be cited independently in the future. For instructions see: http://journals.plos.org/plosone/s/submission-guidelines#loc-laboratory-protocols

We look forward to receiving your revised manuscript.

Kind regards,

Efthimios M. C. Skoulakis, PhD

Academic Editor

PLOS ONE

Reviewers' comments:

Reviewer's Responses to Questions

**Comments to the Author**

1. If the authors have adequately addressed your comments raised in a previous round of review and you feel that this manuscript is now acceptable for publication, you may indicate that here to bypass the “Comments to the Author” section, enter your conflict of interest statement in the “Confidential to Editor” section, and submit your "Accept" recommendation.

Reviewer #1: All comments have been addressed

Reviewer #2: (No Response)

2. Is the manuscript technically sound, and do the data support the conclusions?

Reviewer #1: Yes

Reviewer #2: No

3. Has the statistical analysis been performed appropriately and rigorously? 

Reviewer #1: Yes

Reviewer #2: No

4. Have the authors made all data underlying the findings in their manuscript fully available?

Reviewer #1: Yes

Reviewer #2: No

5. Is the manuscript presented in an intelligible fashion and written in standard English?

Reviewer #1: Yes

Reviewer #2: Yes

6. Review Comments to the Author

Reviewer #1: All questions were addressed by the authors. This is a useful manuscript uncovering limitations of a technique used to predict death in flies. Conceptually, intestinal integrity decay during aging may not be the most common pathway to death in flies.

Reviewer #2: The authors have largely addressed the comments of the reviewers and provided the technical details required for one to replicate the experiments, however, it remains very concerning that the authors still do not provide critical information to properly review this manuscript (max longevity and log rank p values). It is especially troubling that the authors still do not provide access to the data and respond with “we will deposit the raw data in DRYAD upon acceptance of the paper”. The review of scientific data must be done before an article is published and should not be provided upon acceptance of the paper.

It is now clear that none of the experiments have been experimentally replicated and although the statistical analysis allows for “substantial replicate” it is not sufficient to ensure that the results presented are reproducible. The authors responded that “We did no analyses of the median or maximum longevity” yet states in the article: ”there were significant differences in mean longevity between all populations and their controls except CO and s93 (p=0.16”. For a given population, only one measurement (only one experimental replicate) is made with less than 30 individuals (per gender). The statistical validation of the experiment does not mean that the experiment is reproducible and certainly does not warrant that the resulting interpretation is biologically correct. The need for independent experimental replicate of those experiments is especially needed when the ACO longevity is obviously different between the tapping experiment (53-55 days) and the dye experiment (40 days). The lack of experimental replicate also applies to the tapping experiment that the authors have now clearly stated that they did it “to see if our methods for checking for Smurf flies would affect the mean longevity of the fly”. In this case the authors also uses an inappropriate method (t test). The authors still do not justify why they only tested the effect of taping only on the ACO population when the data show obvious differences in longevity of controls (and differences between control and dye treatments are not always significant). It cannot be concluded that the absence of effect on longevity by tapping of the ACO individuals would show that the taping has no effect on the longevity of the other populations.

In the absence of experimental replicates, I am not confident that the data presented are reproducible and that the biological conclusions are scientifically valid and therefore I cannot recommend to accept this manuscript for publication.

7. PLOS authors have the option to publish the peer review history of their article (what does this mean?). If published, this will include your full peer review and any attached files.

Reviewer #1: No

Reviewer #2: Yes: Laurent Seroude

---

## [Author Response · Author response to Decision Letter 2]

3 Mar 2020

Below we reproduce the comments of each referee in normal font and our responses are shown in italic font in the attached document.

Reviewer #1

All questions were addressed by the authors. This is a useful manuscript uncovering limitations of a technique used to predict death in flies. Conceptually, intestinal integrity decay during aging may not be the most common pathway to death in flies.

We appreciate the referee’s comments and would not anticipate that any of our new edits and material would change his/her mind.

Reviewer #2

The authors have largely addressed the comments of the reviewers and provided the technical details required for one to replicate the experiments, however, it remains very concerning that the authors still do not provide critical information to properly review this manuscript (max longevity and log rank p values). It is especially troubling that the authors still do not provide access to the data and respond with “we will deposit the raw data in DRYAD upon acceptance of the paper”. The review of scientific data must be done before an article is published and should not be provided upon acceptance of the paper.

We have now provided in the supplementary section of the manuscript, the maximum longevity by populations and dye treatment, the median longevity by population and dye treatment and the log rank p values by population and dye treatment. The log rank p values actually reveal more significant differences than do the comparisons of mean longevity. In addition, with these revisions we have submitted a separate file with the raw data from the main experiment and from a preliminary experiment. When the paper is accepted for publication, we still intend to place the raw data on the Dryad website.

It is now clear that none of the experiments have been experimentally replicated and although the statistical analysis allows for “substantial replicate” it is not sufficient to ensure that the results presented are reproducible. 

In this revision we have included the results from a preliminary study of the ACO population and dye #1. This preliminary study supports our most important conclusions, (i) the majority of flies never become Smurfs, (ii) those that do become Smurfs typically do so just before death. 

The authors responded that “We did no analyses of the median or maximum longevity” yet states in the article: ”there were significant differences in mean longevity between all populations and their controls except CO and s93 (p=0.16”. For a given population, only one measurement (only one experimental replicate) is made with less than 30 individuals (per gender). 

While it is the case that our experiment consisted of only one experiment with a particular population and dye this observation ignores the results of each population that was tested multiple times with six different dyes. This is important since each dye has the same general effect, (i) a minority of flies becoming Smurfs, (ii) those that do become Smurfs do so shortly before death, (iii) longevity is reduced. So, in our assessment, and we think for many others, the broad consistency and replication of results (i-iii) over different specific dyes and different populations is the important and robust message. These conclusions draw their strength precisely because they have been replicated in so many different situations.

The statistical validation of the experiment does not mean that the experiment is reproducible and certainly does not warrant that the resulting interpretation is biologically correct. The need for independent experimental replicate of those experiments is especially needed when the ACO longevity is obviously different between the tapping experiment (53-55 days) and the dye experiment (40 days). 

By these comments it is apparent the referee is most concerned with replication at different calendar times. We have already alluded to the fact that our main conclusions are supported by replication of different populations and dyes. We also have alluded that the preliminary study, while not replicating the entire experiment, was done at a different calendar date and did give results consistent with the main experiment. The referee points to the different absolute longevities for ACO flies in the tapping experiment and the main experiment. Firstly, for assessing whether tapping effects longevity, the absolute longevity differences do not interfere with our ability to draw conclusions. Secondly, we suspect the referee is making a more subtle suggestion that measurement can change in unpredictable ways in our lab when done at different times. Although I don’t think it is important to understand why longevity might change from one experiment to another, these block effects are observed often in populations we do repeated longevity assays on, and there are several explanations. One that is specific to this experiment is the fact that flies were kept individually in straws until death. However, prior to being placed in straws they were in cultures with many flies of both sexes and some were old enough to mate. However, once placed in straws they could no longer mate and would thus eventually experience a reduction in mortality from the absence of mating (Partridge et al., 1986, J. Insect Physio. 32:925). The relative impact of this reduction in mating would certainly be a function of how many of the flies had mated prior to their isolation in the straws and this could have certainly varied from one experiment to another.

The lack of experimental replicate also applies to the tapping experiment that the authors have now clearly stated that they did it “to see if our methods for checking for Smurf flies would affect the mean longevity of the fly”. In this case the authors also uses an inappropriate method (t test). The authors still do not justify why they only tested the effect of taping only on the ACO population when the data show obvious differences in longevity of controls (and differences between control and dye treatments are not always significant). It cannot be concluded that the absence of effect on longevity by tapping of the ACO individuals would show that the taping has no effect on the longevity of the other populations.

The referee states that a t-test on the means of two large samples of independent observations is inappropriate although the referee provides no justification for that conclusion. I will add our justification. It is the central limit theorem which can be used to show the asymptotic normal distribution of the sample mean, which is independent of the underlying random variables distribution. Since our concern was with the mean longevity decline observed when flies were cultured with dyes testing the effect of tapping on the mean longevity is precisely what should have been done. We understand that by not conducting a tapping experiment under all conditions there is the possibility of alternative outcomes. Thus, we have added the following statement to our “Conclusions” section: “We only tested one population, ACO, for an effect of tapping on mean longevity. Thus, for the ACO population it is clear that dyes are responsible for their reduced longevity not tapping. While it is theoretically possible that the other populations are not affected by the dyes but are affected by tapping, we believe this is an unlikely possibility.”

In the absence of experimental replicates, I am not confident that the data presented are reproducible and that the biological conclusions are scientifically valid and therefore I cannot recommend to accept this manuscript for publication.

As stated, before we believe that there is in fact replication that provides a consistent picture of the effect of dyes on producing the Smurf phenotype. We believe this is an important contribution since there has been no published study with this level of detail about this potentially important technique.

---

## [Decision Letter · Decision Letter 3]

13 Mar 2020

Predicting Death by the Loss of Intestinal Function

PONE-D-19-29500R3

Dear Dr. Bitner,

We are pleased to inform you that your manuscript has been judged scientifically suitable for publication and will be formally accepted for publication once it complies with all outstanding technical requirements.

With kind regards,

Efthimios M. C. Skoulakis, PhD

Academic Editor

PLOS ONE

Additional Editor Comments (optional):

Reviewers' comments:

Reviewer's Responses to Questions

**Comments to the Author**

1. If the authors have adequately addressed your comments raised in a previous round of review and you feel that this manuscript is now acceptable for publication, you may indicate that here to bypass the “Comments to the Author” section, enter your conflict of interest statement in the “Confidential to Editor” section, and submit your "Accept" recommendation.

Reviewer #1: All comments have been addressed

2. Is the manuscript technically sound, and do the data support the conclusions?

Reviewer #1: Yes

3. Has the statistical analysis been performed appropriately and rigorously? 

Reviewer #1: Yes

4. Have the authors made all data underlying the findings in their manuscript fully available?

Reviewer #1: Yes

5. Is the manuscript presented in an intelligible fashion and written in standard English?

Reviewer #1: Yes

6. Review Comments to the Author

Reviewer #1: The authors have provided additional analysis of their data that confirm the main findings. Therefore I believe that the manuscript should be accepted for publication.

7. PLOS authors have the option to publish the peer review history of their article (what does this mean?). If published, this will include your full peer review and any attached files.

Reviewer #1: No

---

## [Editor Report · Acceptance letter]

31 Mar 2020

PONE-D-19-29500R3 

Predicting Death by the Loss of Intestinal Function 

Dear Dr. Bitner:

I am pleased to inform you that your manuscript has been deemed suitable for publication in PLOS ONE. Congratulations! Your manuscript is now with our production department. 

With kind regards,

on behalf of

Dr. Efthimios M. C. Skoulakis 

Academic Editor

PLOS ONE